# Electrophysiological Differentiation of the Effects of Stress and Accent on Lexical Integration in Highly Fluent Bilinguals

**DOI:** 10.3390/brainsci10020113

**Published:** 2020-02-20

**Authors:** Jennifer Lewendon, Anouschka Foltz, Guillaume Thierry

**Affiliations:** 1School of Languages, Literatures and Linguistics, Bangor University, College Rd, Bangor, Wales LL57 2DG, UK; 2Institute of English Studies, University of Graz, Heinrichstraße 36/II, 8010 Graz, Austria; anouschka.foltz@uni-graz.at; 3School of Psychology, Bangor University, Penrallt Rd, Bangor, Wales LL57 2AS, UK; g.thierry@bangor.ac.uk

**Keywords:** lexical stress, bilingualism, event-related brain potentials, word comprehension, implicit priming, speech processing, lexical access

## Abstract

Individuals who acquire a second language (L2) after infancy often retain features of their native language (L1) accent. Cross-language priming studies have shown negative effects of L1 accent on L2 comprehension, but the role of specific speech features, such as lexical stress, is mostly unknown. Here, we investigate whether lexical stress and accent differently modulate semantic processing and cross-language lexical activation in Welsh–English bilinguals, given that English and Welsh differ substantially in terms of stress realisation. In an L2 cross-modal priming paradigm, we manipulated the stress pattern and accent of spoken primes, whilst participants made semantic relatedness judgments on visual word targets. Event-related brain potentials revealed a main effect of stress on target integration, such that stimuli with stress patterns compatible with either the L1 or L2 required less processing effort than stimuli with stress incompatible with both Welsh and English. An independent cross-language phonological overlap manipulation revealed an interaction between accent and L1 access. Interestingly, although it increased processing effort, incorrect stress did not significantly modulate semantic priming effects or covert access to L1 phonological representations. Our results are consistent with the concept of language-specific stress templates, and suggest that accent and lexical stress affect speech comprehension mechanisms differentially.

## 1. Introduction

Bilinguals are often detectable by their native accent. Even a highly fluent, native-like speaker of a second language (L2) will often produce L2 speech with a number of native (L1) phonological and prosodic features [1,2,3]. This foreign accent in L2 speech is thought to result from an interaction between the segmental and suprasegmental characteristics of the native and second languages [4]. The influence that the presence of native features has on second language processing has been the subject of research since the 1960s [5,6,7,8,9]. Lagrou, Hartsuiker and Duyck [9] investigated the influence of L1 accent, alongside sentence context and semantic constraints, on L2 intelligibility and language non-selective access. The experiment attempted to explain the effect of L1 accented L2 speech by bringing together two types of observation: (a) evidence that L1 accent can negatively affect L2 comprehension [8,9,10,11], and (b) the fact that when bilinguals process their second language, they automatically and implicitly activate native language translation representations [12,13,14]. In the experiment, Dutch–English bilinguals made lexical decisions regarding the last word of English sentences produced by a native speaker of English or Dutch. A main effect of interference was found for interlingual homophones (e.g., *lief* “sweet” – *leaf*) with response times significantly longer relative to control stimuli, suggesting that bilinguals encountered interference from their L1 when processing the L2. Furthermore, the homophone interference effect was modulated by accent, such that interference was larger for Dutch-accented English sentences compared to English-accented sentences. Whilst the interaction between accent and homophone interference suggests that native accent modulates language non-selective access, a main effect of accent, with slower responses to the Dutch speaker than the English speaker, suggested that L1 accented speech interfered with processing. The latter result led the authors to suggest that future research should investigate whether L1 accent slows down L2 processing because L1 accent increases L1 salience (and thus interferes with L2 processing), or because L1 accent results in an overall decrease in intelligibility of L2 speech.

Whilst the results of Lagrou, Hartsuiker and Duyck [8,9] suggest that L1 accent in second language comprehension can influence lexical access in bilinguals, the presence of individual sub-phonemic language-specific cues may influence L1 salience in an L2 context [15], and consequently it is not clear which aspects of an L1 accent influence lexical access and in which ways they do so. Foreign accent is not a simple construct, but a “complex of interlingual or idiosyncratic phonological, prosodic and paralinguistic systems” [16] (p. 167). The realisation of L2 intonation and prosody, for example, has been shown to be particularly influenced by corresponding properties of the L1 [17,18,19]. Thus, while Lagrou et al. [8,9] suggest that L1 accent may decrease intelligibility of L2 speech, or increase L1 activation, it is important to consider that accent entails a host of segmental characteristics (e.g., realization of phonemes) and suprasegmental features (e.g., prosody, including intonation, timing and stress) [17,18]. 

One such feature that contributes towards accent is lexical stress, defined as the perceptual salience of a syllable in a word [20]. Consistent with the fact that L1 accent transfers from the first language to the second, L1 stress patterns have been shown to influence the production and comprehension of L2 lexical stress [21,22,23,24]. Lexical stress varies in its realisation both between and within languages. In stress-variable languages it can fundamentally alter the perception of words. For example, in certain languages, lexical stress can be contrastive and mark the difference between the phonology of words which are otherwise segmentally identical (e.g., in English, *insight* vs. *incite*). In contrast, in other languages it is fixed, occurring consistently on specific syllable *loci*. Although speakers of languages without contrastive lexical stress (e.g., French, Hungarian, Welsh) often experience difficulty consciously discriminating between the stress patterns of languages with variable stress [25,26], a host of EEG experiments have revealed that such ‘stress-deaf’ individuals still shown brain sensitivity to stress violations, and in particular to violations of native fixed stress patterns [27,28,29,30,31]. 

Two languages with such contrasting stress systems are English and Welsh. English lexical stress is variable and, despite generally conforming to certain rules, may occur on any syllable within a given word. Stress is indicated for the most part through vowel reduction (e.g., the first syllable of the noun *conflict* is stressed and, depending on the dialect, contains the full vowel [ɒ] or [ɑː], whereas the first syllable of the verb *conflict* is unstressed and contains the reduced vowel [ə]), and/or a combination of suprasegmental features including an increase in pitch, duration and amplitude of the stressed syllable [20,32,33,34]. In contrast, lexical stress in Welsh is highly regular, consistently occurring on the penultimate syllable. Although irregular stress does occur, this is a highly uncommon exception predominantly found in English loanwords [35]. In comparison to English, the relative paucity of research on Welsh lexical stress means that the intricacies of its realisation and perception are less well understood, and consequently remain subject to some debate. Although the current understanding of Welsh lexical stress is incomplete, it appears to be realised on the basis of two key characteristics: shorter duration and lower pitch of the stressed penultimate syllable relative to the unstressed ultima. Thus, contrary to that of the majority of European languages [36], in which the stressed syllable is generally characterised by higher pitch, and greater duration, loudness and salience of the vowel, Welsh stress features phonological prominence of the final unstressed syllable relative to the stressed penult [37]. Despite the long-term language contact situation in Wales, in which Welsh, whilst increasingly spoken as a native language, coexists alongside English with Welsh monolingualism existing solely in some pre-school children, evidence suggests that stress realisation in Welsh has not entirely converged to resemble that of English [35].

Considering the findings of Lagrou et al. [8,9] that suggest L1 accent may either reduce L2 intelligibility or heighten activation of a bilingual’s L1, alongside research suggesting that L1 stress patterns may influence the production and comprehension of L2 lexical stress [21,22,23,24,38], the question thus arises to what extent lexical stress, as an individual suprasegmental feature, contributes to these accent effects. In the present study, we therefore manipulated lexical stress patterns and native speaker accent in a cross-modal priming paradigm (Figure 1) to investigate how stress and accent may differentially affect lexical access. 

We asked highly-fluent Welsh–English bilingual participants to make semantic relatedness judgments on English word pairs. To determine whether accent influences both intelligibility and native-language activation while processing the second language, trisyllabic English auditory prime words were produced either by an L1 English speaker (participants’ L2) or an L1 Welsh speaker (participants’ L1). Furthermore, to determine whether stress placement influences intelligibility and native-language activation, both the L1 English and Welsh speakers produced auditory primes so as to feature stress on the 1st, the 2nd (penult) or the 3rd (ultima) syllable. For all primes, first syllable stress was consistently correct in the sense that it corresponded to natural productions of these English words. In contrast, second and third syllable stress were anomalous in English in the sense that these words did not naturally carry primary stress on the second or third syllables. However, while anomalous in English, the phonetic and phonological prominence of third syllable English stress best approximated Welsh penultimate stress, which is operationalized through increased duration and high pitch on the following (ultima) syllable. Critically, the Welsh translation equivalents of the primes were also trisyllabic words. Thus, when English primes were produced with third syllable stress, their stress patterns matched the fixed penultimate stress patterns of Welsh, and specifically that of their Welsh translation equivalents.

To test the effects of accent and lexical stress placement on L2 intelligibility and L1 salience, we ran two experiments. In Experiment 1, we explored the effects of accent and stress on semantic priming (left panel of Figure 1), with priming indexing intelligibility. In Experiment 2 we tested how accent and stress affect implicit L1 access during L2 processing by measuring implicit phonological priming through the L1 (right panel of Figure 1). As the results of Lagrou et al. [9] might be explained by both a decrease in intelligibility and an increase in L1 accessibility, the hypotheses of Experiments 1 and 2 that we present below are relatively independent. The two experiments were run together with items from each experiment serving as fillers for the other.

Experiment 1 tested the effects of stress placement and accent on semantic priming. Previous studies have shown that intelligibility affects classic behavioral semantic priming effects (i.e., faster responses to target words preceded by a semantically related prime word compared to an unrelated prime word) in that less intelligible prime words reduce the semantic priming effect [39,40]. If L1 accent reduces intelligibility, as hypothesized by Lagrou et al. [8,9], we would expect that L2 English primes produced with an L1 Welsh accent would be more difficult to process. As such, we would anticipate an interaction between accent and semantic priming, as indexed by increased N400 negativity in the case of related pairs (since unrelated pairs should show no priming effects to begin with). Alternatively, if accent does not reduce intelligibility, we should find no interaction between accent and semantic priming. 

If unnatural stress placement reduces intelligibility, we would expect second syllable stress to show the most reduced semantic priming effect because it represents the least natural pronunciation of the prime stimuli. This is the case because English strongly disfavors stress on adjacent syllables. As such, the first syllable (which would carry the main stress in natural productions) would be clearly unstressed in second-syllable stressed primes, rendering them rather unnatural. In contrast, first-syllable stressed primes represent natural productions and third-syllable stressed primes are likely to carry secondary stress on the first syllable, rendering them more similar to natural productions than second-syllable stressed primes. Furthermore, third-syllable stressed primes also best approximate Welsh stress and are therefore additionally likely to be more intelligible to Welsh–English bilingual participants than second-syllable stressed primes. We would therefore expect an increase in amplitude in the classic N400 time window spanning 350–500 ms [38] for 2nd syllable stress primes relative to 1st and 3rd syllable stress in the case of related pairs. Behaviorally, we predict that second-syllable stressed primes would show longer reaction times and lower accuracy.

Experiment 2 investigated whether lexical stress and accent differentially affect access to L1 representations in an L2 context. To test this, we manipulated phonological overlap in the L1 translation equivalents of L2 English primes and targets, such that certain word pairs featured a word-initial phoneme overlap if translated into Welsh, e.g., *hospital* (‘ysbyty’) – *writing* (‘ysgrifennu’) (see Figure 1). Our manipulation was based on prior research showing that phonological overlap through the L1 results in a priming effect, attributed to unconscious native language activation [12,13]. Importantly, all word pairs in Experiment 2 were semantically unrelated (see Methods). We hypothesised that (i) L1 accent would heighten activation of L1 phonological representations, resulting in an increase in implicit phonological priming, and (ii), if Welsh-approximate stress were to increase native language activation, that 3rd syllable stress (compatible with Welsh) would result in increased phonological priming irrespective of accent. We predicted that such priming effects would manifest as a reduction of event-related potentials (ERP) mean amplitudes between 200–400 ms over centroparietal electrode sites for word pairs with phonological overlap in the L1 (i) with Welsh-accented primes and (ii) with third syllable stress primes. Our time window of interest and the topography selected are in accordance with prior research demonstrating that phonological processing and expectancy influences ERPs within the range of the phonological mapping negativity [41,42,43,44], N250 - P325 [45,46,47,48] and early N400 [12,13,49,50] over centroparietal regions. Consistent with prior studies of implicit phonological priming [12,13], we predicted no effect of either accent, stress or overlap on behavioural measures. 

## 2. Materials and Methods

### 2.1. Participants 

Twenty-one Welsh–English bilinguals (14 females, mean age = 24.3; *SD* = 8.6) with normal or corrected-to-normal vision, no learning disabilities, and self-reported normal hearing participated in the experiment. All participants gave written informed consent before taking part in the experiment (approved by the School of Psychology, Bangor University ethics committee, approval no. 2017-16168). All participants began learning Welsh prior to the age of three at home, and had studied through the medium of Welsh up to the age of 12. Age of acquisition for English varied, although only participants who had learnt English either as a second language through formal school tuition, or subsequent to Welsh in a bilingual home were included. For participants who had learnt English formally as a second language at school, tuition did not begin prior to the age of six. All participants except one were right-handed. Table 1 shows participants’ language background for the L1 (Welsh) and L2 (English).

### 2.2. Materials 

Each participant experienced 936 trials across two experimental sessions. Auditory word primes were 39 different trisyllabic English words digitally recorded in English by both a female native English speaker and a female native Welsh speaker at a sampling rate of 48.8 kHz and resampled using Audacity to 44.1 kHz to ensure compatibility with the E-Prime stimulus presentation software. The English speaker selected was a monolingual with a Standard Southern English accent, whilst the Welsh speaker was a non-native English speaker, with a regional accent typical of the Llŷn Peninsula in North Wales. The Welsh speaker was selected from this area as it is a notably Welsh-dominant area. Consequently, whilst the population in Wales consists of a large number of native speakers of English with Welsh accents, it would be atypical to find a native English speaker with an accent characteristic of this region. Each prime word was recorded with stress on the first, second and third syllable in both a Welsh and an English accent, creating six contrasting recordings for each prime word, for a total of 234 prime word recordings (each encountered four times overall during the study). During prime recording, the speakers were initially instructed to practice stress manipulations by changing pitch, duration and loudness of each syllable, whilst producing the same vowels in each case. Inspection of recordings was conducted syllable-by-syllable to ensure no vowel reduction could be auditorily perceived.

Table 2 and Table 3 show the duration, intensity and pitch of each syllable (1st syllable, 2nd syllable, 3rd syllable) produced in each stress condition for the native English speaker and the native Welsh speaker, respectively. Table 4 shows that, as expected, the native English speaker produces the stressed syllable with numerically (and in most cases significantly) longer duration, higher intensity and higher pitch compared to the unstressed syllables across all stress conditions. Interestingly, the native Welsh speaker approximates L1 English stress in certain aspects of all three measured parameters across stress conditions. Specifically, the native Welsh speaker shows native-like duration patterns, such that the stressed syllable across all three stress conditions is numerically (and in most cases significantly) longer compared to the unstressed syllables. Intensity is highest for the stressed syllable in the 2nd and 3rd syllable stress conditions, but not the 1st syllable stress condition. Finally, whilst 1st syllable stress approximates that of the native English speaker’s productions, 2nd and 3rd syllable stress both show overall declining pitch with numerically the highest pitch on the first syllable across all stress conditions. Overall, this suggests that the native Welsh speaker implements English stress similarly to native English speakers in terms of duration, but may not fully implement English-like stress in terms of intensity or pitch. 

Visual word targets were two lists of 117 different words of English varying in length from 2 to 4 syllables, for a total of 234 different target words. Whilst the same auditory primes were used in both the semantic relatedness and phonological overlap conditions (with each individual recording experienced four times overall), two discrete target lists were used in order to manipulate phonological overlap and semantic relatedness separately. That is, one list provided the target words for the phonological overlap manipulation and the other for the semantic relatedness manipulation. Prime and target words were paired to form experimental conditions as follows: (1) Semantic relationship (related condition), as in *hospital – sick* (in Welsh: *ysbyty - gwael*); (2) No semantic relationship (unrelated condition) using target stimuli from the same list as condition (1), as in *hospital* – *publish* (in Welsh: *ysbyty* – *cyhoeddi*), (3) Phonological overlap via Welsh translation (overlap condition), as in *hospital* – *writing* (in Welsh: *ysbyty* – *ysgrifennu*); and (4) No overlap through Welsh (no overlap condition) using target stimuli from the same list as condition (3), as in *hospital* – *rock* (in Welsh: *ysbyty* – *craig*). Critically, target words were rotated across conditions (1) and (2) on the one hand and across conditions (3) and (4) on the other, meaning that all targets featured in the semantically related condition also featured in the unrelated condition and all targets in the phonological overlap condition also featured as targets in the no overlap condition. 

In the critical manipulation for Experiment 1 (related vs. unrelated contrast), the prime and target were either semantically related or unrelated. For all semantically unrelated conditions (2–4), there were no listed associations between prime and target pairs in either the Edinburgh Associative Thesaurus [51] or the University of South Florida Free Association Norms [52] (mean = 0, *SD* = 0), whilst the semantic relationship condition (1) featured a greater degree of associations (mean = 3.0, *SD* = 8.9). In the critical manipulation for Experiment 2 (overlap condition vs. no overlap condition), the L1 translations of prime and target words overlapped or did not overlap in their onset phonemes. A word-initial phonological overlap was selected consistent both with the prediction of the Cohort Model [53,54], namely that word candidate activation occurs within the first 150–200 ms of auditory input (roughly corresponding to the first 1-2 phonemes of a word), and with prior research demonstrating ease of processing for L2 words sharing initial consonants with L1 translations equivalents [14]. 

Each auditory prime word was paired with three possible visual word targets in order to display a different target for each of the three stress recordings, resulting in 117 (3 × 39) prime-target combinations per condition and speaker (117 × 4 conditions × 2 speakers = 936 trials overall). To minimise effects of familiarity, lexical frequency, word length, and concreteness, all words were familiar and had mid-range lexical frequency. Primes and targets were matched across conditions for lexical concreteness and frequency. Frequency measures for English materials were calculated from SUBTLEX [55] (mean = 4.64, *SD* = 0.70). An analogous corpus is not available for Welsh, so frequency measures for Welsh materials were calculated from Cronfa Electroneg o Gymraeg (CEG; [56]) (mean = 1.97, *SD* = 0.65). Concreteness measures for English materials were calculated from the *Concreteness ratings for 40 thousand generally known English word lemmas* corpus [57] (mean = 3.55, *SD =* 1.02), and, due to corpus unavailability, assumed to be similar for Welsh translations. 

### 2.3. Procedure

Participants were tested in two sessions separated by at least a day. Half of the participants were exposed to the Welsh-accented stimuli during their first session, whilst the other half heard English-accented stimuli first. Each testing session consisted of 468 trials, 234 forming the semantic relatedness paradigm and the remaining 234 forming the implicit phonological priming paradigm. A trial began with a fixation cross presented for the duration of 100 ms on a 17” CRT monitor at a distance of 100 cm from the participant’s eyes. The fixation cross was followed by an auditory prime, which was played over loudspeakers set around the monitor. Following the auditory prime, a second fixation cross was displayed for a variable ISI of 160–240 ms. This was followed by the visual target word, which was presented in black Times New Roman font, size 14 points on a light grey background. Participants were instructed to indicate whether prime and target pairs were semantically related by pressing a button within a 2000 ms response window, and response-hand side was counterbalanced across participants. Participants’ response immediately triggered the next trial. Prior to commencing the full experiment, participants underwent a brief training period to ensure they were familiar with the procedure. 

### 2.4. Data Analysis

#### 2.4.1. ERP Recording and Pre-Processing

EEG data were recorded at 2048 Hz using a BioSemi system with 128 active Ag/AgCl electrodes with the passive common mode sense (CMS) electrode as reference and driven right leg (DRL) as ground. Prior to recording, a cap was fitted to secure the EEG electrodes in place, and electrode impedances were reduced to < 5 kΩ. Six further facial bipolar electrodes positioned on the outer canthi of each eye and in the inferior and superior areas of the left and right orbits provided bipolar recordings of the horizontal and vertical electrooculograms (EOG). Participants were instructed to blink and make repeated vertical and horizontal eye movements during an EEG recording prior to the main experiments in order to acquire eye-movement data for subsequent correction. Data were resampled to 1024 Hz prior to analysis, re-referenced offline to the global average reference and filtered offline using a 30 Hz (48 dB/oct) low-pass and 0.1 Hz (12 dB/oct) high-pass Butterworth Zero Phase shift band-pass filter. Noisy electrodes were replaced on an individual basis by means of spherical interpolation, with interpolated electrodes consisting of less than 5% of the data, and for each condition there were a minimum of 30 trials per participant. Ocular correction was conducted using Independent Component Analysis (ICA) following visual inspection of the data using the AMICA procedure [58]. Data were then segmented into epochs ranging from −200 to 1000 ms relative to the onset of the visual target word, and baseline correction was performed relative to 200 ms pre-stimulus activity. 

#### 2.4.2. Modelling of Behavioural Data

For both experiments, reaction time data (RT) were log transformed so as to be normally distributed and analysed via a linear mixed effect model (*lmer* function in lme4). Fixed effects were centred to minimise collinearity, and random effects, including prime and participant intercepts and slopes were modelled and systematically trimmed such that interactions were removed until the model converged [59]. Subsequently, fixed and random effects and interactions that did not significantly contribute to model fit were systematically removed from the initial model. Accuracy data were submitted to generalized mixed-effects modelling (*glmer* with a binomial link function in the lme4 v1.12 library [60]), after centring fixed factors to minimise collinearity. As in the reaction time analysis, random effects including participant and item intercepts and slopes were modelled and systematically trimmed until the model converged, and fixed and random effects and interactions that did not significantly contribute to model fit were systematically removed. 

#### 2.4.3. ERP Analysis

In Experiment 1, mean ERP amplitudes were analysed in an epoch corresponding to the classic N400 window (350-500 ms) in which semantic priming is most observable over the central scalp regions [61] to determine whether prime stress influenced semantic integration processes. In Experiment 2, mean ERP amplitudes were analysed between 200 and 400 ms, consistent with prior research demonstrating that phonological processing and expectancy influences ERPs within the range of the phonological mapping negativity [41,42,43,44], N250 - P325 [45,46,47,48] and early N400 [12,13,49,50] over centroparietal regions.

ERP data were analysed by means of two repeated-measures analysis of variance (ANOVA), one for semantic relatedness (Experiment 1) and one for cross-language phonological priming (Experiment 2). In the case of the semantic relatedness manipulation, mean amplitudes for all time windows were analysed over 14 central electrodes where the N400 is usually maximal with accent (English, Welsh), prime stress (syllable 1, 2 or 3), and relatedness (related, unrelated) as independent variables. For the phonological priming analysis, the repeated measures ANOVA was conducted over 12 centroparietal electrodes with accent (Welsh, English), overlap in L1 (overlap, no overlap), and prime stress (syllable 1, 2 or 3) as factors. 

## 3. Results

### 3.1. Experiment 1: Semantic Priming

#### 3.1.1. Behavioural Results

Figure 2 shows an overview of the RT and accuracy results for Experiment 1. RTs were modelled as a function of the three within-subject factors, accent (Welsh, English), prime stress (syllable 1, 2 or 3) and semantic relatedness (related, unrelated). Accent and stress fixed effects did not significantly contribute to model fit and were removed. Results revealed a main effect of semantic relatedness, with unrelated pairs responded to significantly faster than related pairs (*b* = −0.029, *SE* = <0.009, *t* = −3.22, *p =* 0.002). Accuracy data were submitted to generalized mixed-effects modelling, but the model failed to converge with all fixed effects included. Instead, data were analysed separately by accent. For Welsh-accented prime words the fixed effect of stress did not significantly contribute to model fit and was removed from the model. There was a significant effect of relatedness (*b* = 1.376, *SE* = 0.287, *z* = 4.78, *p* < 0.001), such that responses to unrelated stimuli were significantly more accurate than to related stimuli. For the English-accented analysis, the final model failed to converge and the random effects structure was consequently simplified until convergence was achieved. Simplification of the random effects structure did not affect the results of the model. The results revealed no effect of stress (*b* = 0.057, *SE* = 0.051, *z* = 1.12, *p =* 0.259), but a significant main effect of relatedness (*b* = 1.574, *SE* = 0.269, *z* = 5.84, *p* <0.001), such that accuracy for unrelated word pairs was again significantly higher than for related pairs, and a significant relatedness by stress interaction (*b* = 0.138, *SE* = <0.051, *z* = 2.70, *p =* 0.006). Post hoc tests found no effects of stress on accuracy for either unrelated (*b* = 0.115, *SE* = 0.160, *z* = 0.72, *p =* 0.471) or related stimuli (*b* = 0.080, *SE* = 0.047, *z* = −1.70, *p =* 0.080), although the latter just failed to reach significance.

#### 3.1.2. Electrophysiological Results

A repeated measures ANOVA on ERP mean amplitudes in the 350–500 ms time window revealed a main effect of relatedness (F(1, 18) = 19.80, *p* < 0.001, η_p_^2^ = 0.524) such that N400 amplitude was significantly more negative in the unrelated than the related condition for all stress and accent conditions (Figure 3). There was no significant main effect of accent (F(1, 18) = 0.06, *p* = 0.802, η_p_^2^ = 0.004), but the main effect of stress was marginal (F(2, 36) = 2.84, *p* = 0.07, η_p_^2^ = 0.137). Explorative post hoc comparisons of the three stress conditions showed that 2nd syllable stress elicited greater ERP amplitudes than 1st syllable stress (t(18) = 2.35, *p =* 0.023) but there was neither a significant difference between 1st and 3rd (t(18) = 0.98, *p =* 0.332), nor between 2nd and 3rd syllable stress (t(18) = −1.39, *p =* 0.172). There was no interaction between relatedness and accent (Figure 3A; F(1, 18) = 0.16, *p* = 0.691, η_p_^2^ = 0.009); relatedness and stress (Figure 3B; F(2, 36) = 0.10, *p* = 0.901, η_p_^2^ = 0.006); or accent and stress (F(2, 36) = 0.54, *p* = 0.582, η_p_^2^ = 0.030) and the three-way interaction was also not significant (F(2, 36) = 0.24, *p* = 0.781, η_p_^2^ = 0.014).

### 3.2. Experiment 2: Cross-Language Phonological Priming

#### 3.2.1. Behavioural Results

Figure 4 shows an overview of the RT and accuracy results for Experiment 2. RTs were modelled for accent (Welsh, English), prime stress (syllable 1, 2 or 3) and L1 overlap (overlap, no overlap), centred to minimise collinearity. Accent, stress and overlap fixed effects did not significantly contribute to model fit and were removed. Therefore, as predicted, accent, stress and lexical overlap had no effect on RTs. 

Accuracy data were submitted to generalized mixed-effects modelling and accent, prime stress and overlap were again centred to minimise collinearity. Random effects were modelled and systematically trimmed but the model failed to converge when all fixed effects were included. Data were analysed similarly to Experiment 1, modelled separately by accent. For Welsh-accented prime words, the fixed effect of overlap did not significantly contribute to model fit and was removed. There was a significant main effect of stress (*b* = 0.239, *SE* = 0.076, *z* = 3.019, *p =* 0.002). Post hoc tests showed that accuracy in the 2nd syllable stress condition (Mean = 92%, *SE* = 5%) was significantly higher than that of the natural 1st syllable stress condition (Mean = 91%, *SE* = 5%, *b* = 0.431, *SE* = 0.180, *z* = −2.40, *p =* 0.043) and this was also true when comparing 3rd to 1st syllable stress (Mean = 93%, *SE* = 5%, *b* = 0.543, *SE* = 0.185, *z* = −2.94, *p =* 0.009). There was no significant difference between 2nd or 3rd syllable stress (*b* = 0.111, *SE* = 0.198, *z* = −0.56, *p =* 0.839). There was no effect of accent, stress or overlap for English accented prime words, with the three fixed effects not significantly contributing to model fit and being removed from the model.

#### 3.2.2. Electrophysiological Results

The repeated measures ANOVA conducted on ERP amplitudes in the 200–400 ms window revealed a significant main effect of stress (F(2, 36) = 5.56, *p* = 0.008, η_p_^2^ = 0.236, Figure 5A). Post hoc analyses showed that ERP mean amplitudes were significantly more negative for target words preceded by 2nd syllable stress as compared to natural 1st syllable stress primes (t(18) = 2.93, *p =* 0.006) and 3rd syllable stress (t(18) = −2.83, *p =* 0.007). There was no significant difference between target words preceded by 1st syllable stress primes relative to 3rd syllable stress primes (t(18) = 0.09, *p =* 0.92). We also found a significant interaction between phonological overlap and accent (F(1, 18) = 5.95, *p* = 0.025, η_p_^2^ = 0.249, Figure 5B). Mean ERP amplitudes for overlapping pairs were significantly less negative than for non-overlapping pairs (t(18) = 2.35, *p* = 0.02) when primes were produced in a Welsh accent, but no such difference was found when primes were produced in an English accent (t(18) = 0.92, *p* = 0.36). Importantly, there was no interaction between phonological overlap and stress (F(2, 36) = 2.02, *p* = 0.693, η_p_^2^ = 0.020, Figure 5B). No other main effect or interaction was significant.

## 4. Discussion

In the current study, we investigated whether lexical stress and accent differently modulate semantic processing (Experiment 1) and cross-language lexical activation (Experiment 2) in highly proficient Welsh–English bilinguals. In Experiment 1, there was no effect of accent, and only a marginal effect of lexical stress on semantic priming as indexed by mean N400 amplitude. In contrast, in Experiment 2 we found an implicit L1 priming effect, but only for Welsh-accented primes, with significantly less negative mean amplitudes for stimuli overlapping through the L1 compared to those not overlapping. We also found a significant main effect of stress, with ERP amplitudes for 2nd syllable stress primes significantly more negative than 1st or 3rd syllable stress primes. The results are discussed for each experiment individually below.

### 4.1. Experiment 1

#### 4.1.1. Behavioural Data

Both the reaction time results, with faster responses to unrelated words than related words, and the accuracy results, with higher accuracy for unrelated compared to related words, differed from those classically reported in semantic priming paradigms [62,63]. We speculate that the increased RTs and decreased accuracy in response to related word pairs may relate to two characteristics of the experimental design: (i) Prime words were repeated 12 times, albeit with three different stress patterns and in two different accents. This may have led participants to generate incorrect expectations about any given repeated prime (e.g., having heard a prime paired with a related target may have resulted in participants expecting any other iteration of the same prime word to be paired with an unrelated target). (ii) Given the design of the study, only 25% of word pairs were semantically related, making related pairs overall infrequent and less expected. This unexpectedness may have increased RTs and decreased accuracy for related pairs. 

#### 4.1.2. Electrophysiological Data

In contrast to behavioural results, N400 modulation showed the expected semantic priming effects, thought to index the spread of activation through the conceptual system [61]. Behavioural results and N400 effects were thus not aligned, as has been shown repeatedly in ERP studies of semantic processing in which behavioural data were recorded [12,64]. This result is consistent with the view that the N400 is mostly insensitive to explicit task requirements or conscious evaluation of the stimuli [61]. 

Experiment 1 tested the proposals put forward by Lagrou et al. [8,9] that L1 accent, when present in the L2, results either in heightened salience of the L1 or overall reduced L2 intelligibility. The results of Experiment 1 showed no effect of accent on reaction times, accuracy or, critically, N400 mean amplitudes. Neither did we find any interaction between accent and semantic relatedness. This suggests that L1 accent in a second language context does not measurably affect intelligibility, since such an effect should have resulted in a modulation of the N400 effect across accent conditions. 

Furthermore, due to the tendency for native-language stress patterns to influence the production and comprehension of L2 lexical stress [21,22,23,24,38], in Experiment 1 we sought to differentiate between the effects of accent and lexical stress on L2 intelligibility. We hypothesized that if unnatural stress placement reduced intelligibility, we would expect second syllable stress, as the least natural pronunciation, to result in the greatest reduction in semantic priming. Furthermore, in addition to being the least natural stress production in English, 2nd syllable stress was also considered to least resemble stress in the participants’ L1 (Welsh), and consequently the least intelligible for the Welsh–English bilingual participants. We hypothesized that this would have resulted in increased N400 amplitude for related words pairs in the 2nd syllable stress condition as compared to both the 1st syllable stress condition (natural stress), and the 3rd syllable stress condition, given that the latter best approximates Welsh stress pattern. Instead of the anticipated stress by relatedness interaction, our results showed a marginal main effect of stress driven by 2nd syllable stress primes. Although marginal, and thus any interpretation should be tentative, the effect suggests that 2nd syllable stress interfered with the processing of visual word targets, irrespective of semantic relatedness or accent. 

### 4.2. Experiment 2

#### 4.2.1. Behavoural Data

As expected, there were no significant effects in the behavioural results for Experiment 2, except for a main effect of stress on accuracy for Welsh-accented primes. This effect is not easy to interpret, because: (i) it was very small in size (maximally 2% accuracy difference); (ii) the full model testing the accent by stress interaction failed to converge; and (iii) we must keep in mind that in Experiment 2, all prime-target word pairs were unrelated in the context of a semantic relatedness judgement task. For these reasons, we refrain from over-interpreting this result. 

#### 4.2.2. Electrophysiological Data

For Experiment 2, the ERP data revealed an interaction between speaker accent and L1 phonological overlap on mean ERP amplitudes between 200 and 400 ms post target onset. For Welsh-accented primes, ERP amplitude was significantly less negative when prime and target words phonologically overlapped through L1 Welsh translations relative to the no overlap condition. We interpret this result as evidence of heightened implicit L1 activation when L2 primes were heard with a native L1 accent. The finding thus sheds light on the two proposals put forward by Lagrou et al. [8,9], namely that L1 accented L2 speech either decreases overall intelligibility or increases salience of the L1. Where Experiment 1 results failed to provide evidence in favour of reduced intelligibility by native accent, the latter result points toward increased L1 salience. Heightened activation of the native language due to the presence of L1 accent bears particular relevance for both current models of bilingual word recognition, and our understanding of cognitive advantage relative to the notion of the ‘bilingual experience’ [65]. Firstly, the finding highlights the necessity for models of bilingual word recognition to give greater weight to suprasegmental information. Certain models (e.g., BLINCS [66]) entirely overlook suprasegmental information whilst others fail to account for such effects even though they acknowledge their relevance in bilingual word recognition. For example, to account for segmental input, the Bilingual Model of Lexical Access (BIMOLA) [67], incorporates a feature, a phoneme and a word layer. Above this, a global language information layer accounts for the influence of top-down information and contextual factors, including language mode. ‘Language mode’, which is considered to be the state of activation of a bilingual’s languages and language-processing mechanisms at any given point in time [68], is a continuum in which more general (language proficiency, attitudes, socioeconomic status) and context-specific (topic, language mixing and language of use) factors influence the level of activation of each language, and such factors are likely to include accent [68].

Recall that, in Experiment 2, we expected a stress by L1 overlap interaction, with Welsh- approximate 3rd syllable stress increasing L1 activation. Instead, ERP results unexpectedly showed a main effect of stress, eliciting greater negativity in the 200–400 ms time-window for 2nd syllable stressed primes relative to 1st and 3rd syllable. Strikingly, this pattern is consistent with the main prediction we made for Experiment 1, namely that Welsh–English bilinguals would struggle processing stress patterns that are anomalous both with regard to the L1 and the L2. It may be considered surprising that 3rd syllable stress appeared easier to process than 2nd syllable stress, given the paucity of its occurrence in trisyllabic English words [69]. However, given that 3rd syllable stress was processed by participants in a manner similar to natural stress, we propose two interpretations for this observation: Firstly, as mentioned above, 2nd syllable stress may differ from natural (1st syllable) and 3rd syllable stress because it sounds least natural. Because English strongly disfavors stress on adjacent syllables, we had assumed that the initial syllable of 2nd syllable stress words would be unstressed, whilst in third-syllable stress primes the initial syllable would be likely to carry secondary stress, rendering them more similar to natural productions. However, when considered in light of the phonetic measures of stress listed in Table 2 and Table 3 this explanation is less likely. With the exception of intensity for the Welsh speaker, all phonetic measures (duration, pitch, intensity) of the 1st syllable stress were consistently reduced for 3rd syllable stress words relative to 1st and 2nd syllable stress words. Thus, for the 3rd syllable stressed words, the presence of secondary stress on the initial syllable is difficult to determine, with the perceptual prominence of syllable 1 actually appearing reduced relative to the other stress conditions. 

Secondly, it is important to consider that the Welsh–English bilingual participants tested were native speakers of a phonologically-fixed stress language. When speakers of languages in which stress is based on phonological rules such as syllabic structure or vocalic peaks (as opposed to morphology) learn their language in infancy, it is thought they are able to establish whether their language features contrastive stress prior to the establishment of a lexicon, that is, pre-lexically [70]. This process of figuring out the importance of stress in the native language seems to influence the degree to which stress is encoded as a feature within the lexicon [24,25,26,27,28,29]. In place of lexical stress patterns encoded into individual lexical entries, it has been proposed that such speakers instead understand their native language stress patterns based upon long-term representations, labelled pre-lexical stress templates [29,71]. As second language English speakers, Welsh–English bilinguals would thus have failed to develop a sensitivity to lexical stress in infancy and they would be unable to incorporate stress information into the lexical entries of subsequently acquired L2 words [24,65]. In other words, these highly fluent bilinguals would understand stress in their L2 English on the basis of established native-language fixed stress templates. This may explain why 3rd syllable stress in L2 English is less difficult to process than 2nd syllable stress, given that it shares a number of features with the fixed stress template of the L1 Welsh. The data collected in the present study cannot definitively tease apart these two interpretations, and future experiments involving stress manipulations in bilinguals will hopefully resolve this question.

## 5. Conclusions

In sum, we sought to determine the degree to which stress and accent differentially affect parallel language activation in bilinguals. Remarkably, we found that stress did not interact with L1 phonological overlap and, by itself, failed to modulate cross-language activation. In contrast, the accent by L1 overlap interaction reported in Experiment 2 supports prior suggestions that native accent may heighten L1 salience in an L2 context [8,9]. The lack of an interaction between stress and accent points to independence between these characteristics of language regarding their respective contribution to cross-language activation and lexical processing in highly fluent bilinguals. Whilst native accent heightens the activation of the non-target native language, L1 approximate lexical stress placement in L2 words appears to have no such effect. Instead, L2 stress patterns congruent with those of the native language seem to be processed with relative ease, an effect possibly deriving from pervasive L1-generated, pre-lexical stress templates.

## Figures and Tables

**Figure 1 brainsci-10-00113-f001:**
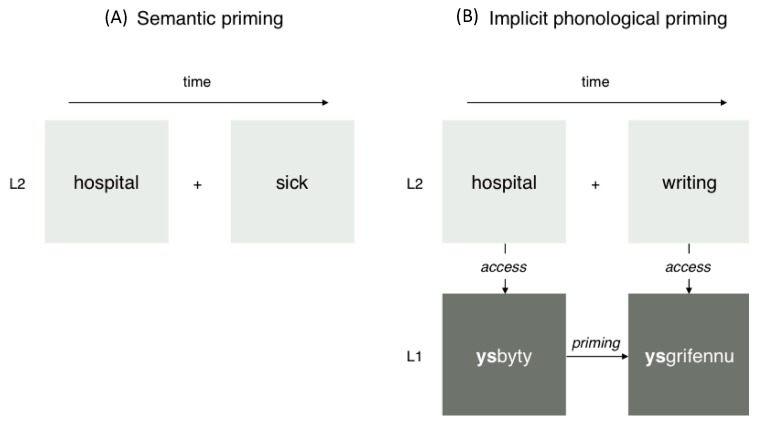
(**A**) Semantic priming paradigm (Experiment 1) and (**B**) implicit phonological priming paradigm (Experiment 2). In both experiments, participants heard an L2 prime word, followed by an L2 visual word target. For the implicit priming paradigm, unconscious access to L1 translations with word-initial phonological overlap would result in implicit priming between otherwise unrelated L2 word pairs.

**Figure 2 brainsci-10-00113-f002:**
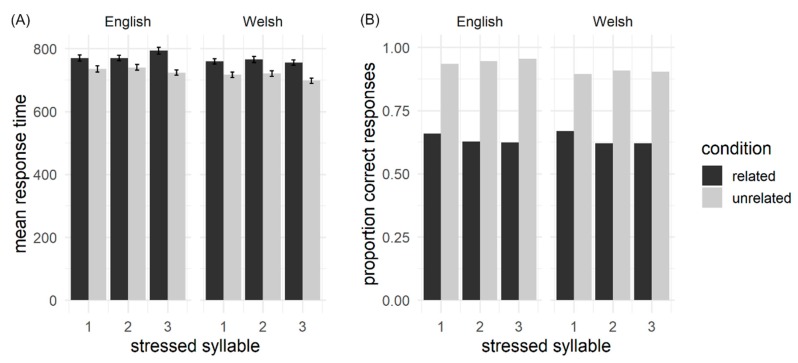
Summary of the behavioural results for Experiment 1. (**A**) reactions times; error bars depict standard error of the mean. (**B**) accuracy.

**Figure 3 brainsci-10-00113-f003:**
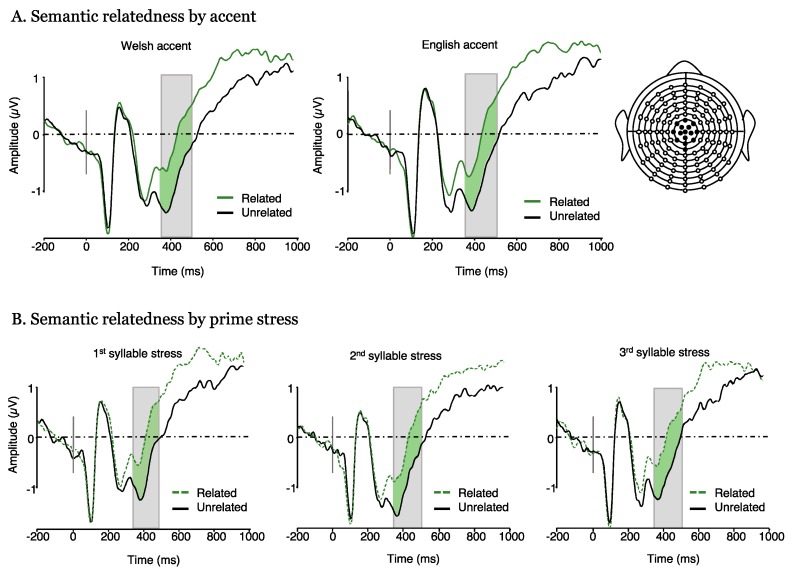
Event-related potential plots from Experiment 1 (semantic priming) plotted for each of the two accents and each of the three stress conditions. (**A**) semantic relatedness effect by accent (no interaction). (**B**) semantic relatedness by stress (no interaction).

**Figure 4 brainsci-10-00113-f004:**
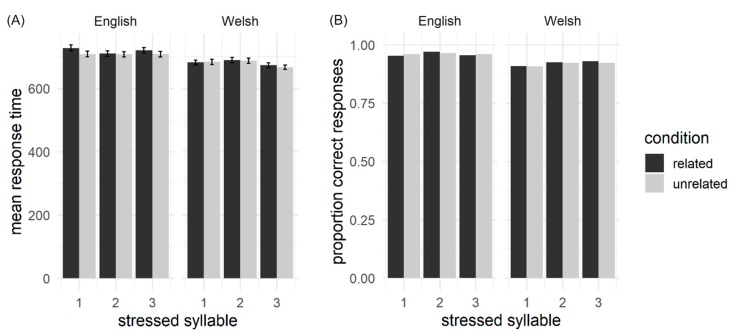
Summary of the behavioural results for Experiment 2. (**A**) reactions times; error bars depict standard error of the mean. (**B**) accuracy.

**Figure 5 brainsci-10-00113-f005:**
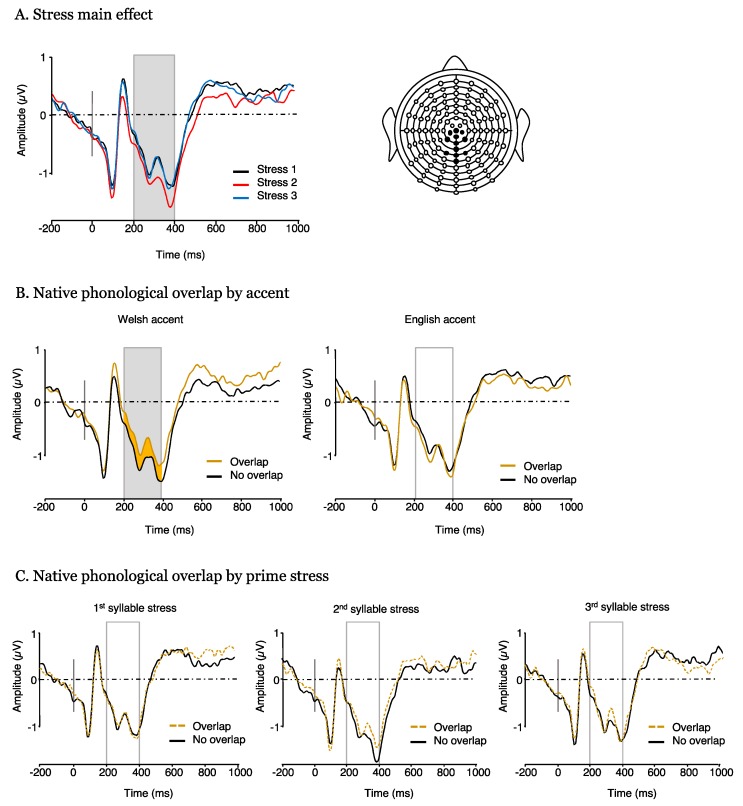
ERP plots obtained in Experiment 2 (cross-language phonological priming). (**A**) main effect of stress across both related and unrelated trials. (**B**) Phonological overlap by accent interaction. (**C**) Phonological overlap by stress (no interaction).

**Table 1 brainsci-10-00113-t001:** Participants’ language background.

Measure	Mean	SD
Age of Welsh acquisition	0.2	0.9
Age of English acquisition	3.8	2.29
Daily Welsh usage (%)	64.5	20.2
Daily English usage (%)	35.2	19.5

**Table 2 brainsci-10-00113-t002:** Phonetic parameters for each stress condition for the native English speaker.

	1st Syllable			2nd Syllable			3rd Syllable		
Stress Cond.	1	2	3	1	2	3	1	2	3
Duration (s)	**0.195**	0.173	0.156	0.158	**0.232**	0.164	0.340	0.354	**0.425**
*SD*	0.056	0.058	0.060	0.050	0.056	0.054	0.117	0.115	0.107
Intensity (dB)	**76.8**	73.3	70.7	73.5	**75.5**	72.5	73.6	73.4	**74.6**
*SD*	1.69	2.46	2.39	2.33	1.87	2.34	2.10	2.61	1.85
Pitch (Hz)	**226.8**	189.8	187.2	184.0	**202.3**	187.2	158.3	157.4	**171.5**
*SD*	17.9	25.7	16.7	12.2	16.5	16.5	5.4	10.5	8.3

Mean values for duration (in seconds), average intensity (in decibel) and average pitch (F0, in Hertz) by syllable, with the numerically highest values for each syllable given in bold face.

**Table 3 brainsci-10-00113-t003:** Phonetic parameters for each stress condition for the native Welsh speaker.

	1st Syllable			2nd Syllable			3rd Syllable		
Stress Cond.	1	2	3	1	2	3	1	2	3
Duration	**0.190**	0.173	0.144	0.132	**0.219**	0.132	0.314	0.333	**0.369**
*SD*	0.053	0.055	0.044	0.044	0.054	0.041	0.093	0.096	0.098
Intensity	72.3	69.6	**73.3**	71.9	**72.6**	71.6	71.5	70.6	**73.4**
*SD*	2.13	2.40	2.75	1.88	1.73	2.05	1.67	1.82	2.19
Pitch	**211.9**	195.4	187.0	**205.9**	205.3	182.9	**224.2**	180.1	172.5
*SD*	11.7	12.7	11.0	16.3	11.0	11.6	24.5	12.9	8.7

Mean values for duration (in seconds), average intensity (in decibel) and average pitch (F0, in Hertz) by syllable, with the numerically highest values for each syllable given in bold face.

**Table 4 brainsci-10-00113-t004:** Statistical comparisons across stress conditions for duration, intensity and pitch. ANOVA result is stated in parenthesis and p-values for post hoc comparisons are listed in the Table.

	English Speaker			Welsh Speaker		
	1st Syllable	2nd Syllable	3rd Syllable	1st Syllable	2nd Syllable	3rd Syllable
Duration	(ANOVA: F = 84.18, p < 0.001)			(ANOVA: F = 67.08, p < 0.001)		
Stress 1–stress 2	0.002	<0.001	0.532	0.452	<0.001	0.136
Stress 1–stress 3	<0.001	1.000	<0.001	<0.001	1.000	<0.001
Stress 2–stress 3	<0.366	<0.001	<0.001	<0.001	<0.001	<0.001
Intensity	(ANOVA: F = 84.404, p < 0.001)			(ANOVA: F = 26.614, p < 0.001		
Stress 1–stress 2	<0.001	<0.001	1.000	<0.001	0.714	0.405
Stress 1–stress 3	<0.001	0.329	0.308	0.405	1.000	<0.001
Stress 2–stress 3	<0.001	<0.001	0.087	<0.001	0.254	<0.001
Pitch	(ANOVA: F = 80.90, p < 0.001)			(ANOVA: F = 41.80, p < 0.001)		
Stress 1–stress 2	<0.001	<0.001	1.000	<0.001	1.000	<0.001
Stress 1–stress 3	<0.001	1.000	0.002	<0.001	<0.001	<0.001
Stress 2–stress 3	1.000	<0.001	<0.001	0.150	<0.001	0.312

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
