# Peer review of "Electrophysiological Differentiation of the Effects of Stress and Accent on Lexical Integration in Highly Fluent Bilinguals"

_brainsci, 2020, doi:10.3390/brainsci10020113_

Round 1
Reviewer 1 Report
This study investigates the priming effects that L1 has on L2, specifically in lexical stress and accent modulate semantic activation in two languages with different stress patterns. The results show that similar stress patterns across langauges facilitate target integration; but stress doesn't have nay priming effects to L1 phonological representations. This study is a nice and clear study which and, honestly, I don't believe there is reason to change it, so I would not suggest any substatial changes. One thing that I didn't get clear is whether in the experimental items each word was recorded three times, each time with a different stress (same words different stress paterns) or there were three kinds of stress word patterns. It suggest to the authors, in order to be more clarifying for the reaters, to include a list of the items as an appendix the used for the study marking the different stresses they have used in each word. It would help the readers to better understand the experimental design. Conclusions The article in the present form is appropiate. It is a nice and well designed studyAuthor Response
Please see the attachment.

Reviewer 2 Report
SUMMARY
The authors report an n = 21 EEG study on the influence of L1 accent and stress on L2 semantic and phonological priming. The authors report that semantic relatedeness of L2 primes slows down processing and decreases accuracy and increases N400 amplitude. The authors also report that stress affected accuracy in the processing of L1-accented phonlogically-primed targets; furthermore, they report that ERPs in a 200–400-ms time window are affected by stress; for L1-accented primes, this ERP was alsop affected by phonological overlap.
I am not an expert on bilingualism so my review is largely restricted to the logical consistency of the motivation and interpretation as well as the soundness of data processing and statistical analysis.
I found the paper hard to read and the writing not straightforward, but this might be solely due to my limited expertise.
MAJOR
4/162: Please provide table where the phonetic parameters comprising stress are listed for the "stress on the first, second, or third syllable" conditions; also split table by accent to show that the prominence of each of the three syllables did not differ between the two speakers.
5/173: How did the authors ensure that there was (no) "semantic relationship" or "phonological overlap" in the according conditions? Did they run some sort of corpus study or a norming experiment? Did they do some calculation of phonetic / phonological distance? Please explain.
6/253 / 6/256: The authors need to provide references for their arbitrary selection of electrodes of interest.
7/270f / 7/277: I have a hard time understanding the statistical analysis here. The authors say (207f) that they split the accuracy analysis by accent, separately analyzing "Welsh-accented" and "English-accented" accuracy data. But then, they mention that the "Welsh-accented" model contained a fixed effect of "accent" and that a "fixed effect of accent" was removed from the model. How can their be a fixed effect of accent in each of the models if the authors split the model by accent? Please check.
MINOR
1/30f: It would be good for the reader to get a definition of "accent", also in contrast with "stress". Currently, it is not fully clear which features are part of "accent" and whether language-specific "stress" is included in "accent".
1/40: "The effect was modulated" Does this mean that the magnitude of the interference effect differed between the L1 Dutch-accented and English sentences? Here, it just says that "sentences were responded to more slowly", which says nothing about the modulation of the interference effect.
2/51: I think the text would be easier to understand if the authors would avoid the term "accentuation" for "stress". If I understand them correctly, they say that "stress" is one feature of "accent"; then, they say that the "stress" is "accentuation". I know what they mean, but maybe it is enough to mention here "perceptual prominence"?
2/70: It would be helpful to read what features are included in "accent".
2/88 / 3/100: I think the authors' description of stress in Welsh requires a bit of cleaning to improve readability. Above, they say that the penulatimate syllable is stressed in Welsh, while the final syllable is phonetically and phonologically more prominent. In short: non-prominent stress on the penultimate in Welsh. Now, the authors say that "third-syllable stress best approximated Welsh penultimate stress". To facilitate understanding, this sentence should really be "the phonetic and phonological prominence of the third syllable best approximated Welsh penultimate stress, which is operationalized through non-prominence." For the reader, there should be a clear difference between prominence and stress: in Welsh, the non-prominent is the stressed, in English, the prominent is the stressed. Please rework.
3/93: I do not understand why "accent", "semantic relatedness", and "phonological overlap" are all being manipulated here to investigate "lexical stress".
3/101: I do not understand how the authors manipulated "accent". They say they manipulated "stress", which I understand. How did they manipulate "accent" in addition?
3/127: How was "accent" manipulated in addition to "stress"? And why? From the entire introduction, it is not fully clear to me why "accent" is of interest here. I thought the manuscript is on transfer effects of native "stress".
4/141: Please provide literature for hypothesized time window and component for phonological priming effect.
6/225: Please provide details on how much data was rejected.
6/250: "data was" should be "data were"
11/392f: Something is odd with the grammar of this sentence, please check.
